# Short communication: Landlab v2.0: A software package for Earth surface dynamics

Katherine R. Barnhart[1, 2, 3], Eric W. H. Hutton[4, 5], Gregory E. Tucker[1, 2, 4], Nicole M. Gasparini[6], Erkan Istanbulluoglu[7], Daniel E. J. Hobley[8], Nathan J. Lyons[6], Margaux Mouchene[9], Sai Siddhartha Nudurupati[7], Jordan M. Adams[10], and Christina Bandaragoda[7]

[1]University of Colorado at Boulder, Cooperative Institute for Research in Environmental Sciences, Boulder, Colorado
[2]University of Colorado at Boulder, Department of Geological Sciences, Boulder, Colorado
[3]Present affiliation: U.S. Geological Survey, Landslide Hazards Program, 1711 Illinois St., Golden, CO
[4]University of Colorado at Boulder, Community Surface Dynamics Modeling System Integration Facility, Boulder, Colorado
[5]University of Colorado at Boulder, Institute for Arctic and Alpine Research, Boulder, Colorado
[6]Tulane University, Department of Earth and Environmental Sciences, New Orleans, Louisiana
[7]University of Washington, Department of Civil and Environmental Engineering, Seattle, Washington
[8]Cardiff University, School of Earth and Ocean Sciences, Cardiff, Wales, United Kingdom
[9]Univ. Grenoble Alpes, INRAE, ETNA, F-38402 St-Martin-d'Hères, France
[10]Delgado Community College, Division of Science and Math, New Orleans, Louisiana

**Correspondence:** Katherine Barnhart (krbarnhart@usgs.gov)

**Abstract.** Numerical simulation of the form and characteristics of Earth's surface provides insight into its evolution. Landlab is an Open Source Python package that contains modularized elements of numerical models for Earth's surface, thus reducing time required for researchers to create new or reimplement existing models. Landlab contains a gridding engine which represents the model domain as a dual graph of structured quadrilaterals (e.g., raster) or irregular Voronoi polygon-Delaunay triangle mesh (e.g., regular hexagons, radially symmetric meshes, fully irregular meshes). Landlab also contains *components*—modular implementations of single physical processes—and a suite of utilities that support numerical methods, input/output, and visualization. This contribution describes package development since version 1.0 and backward-compatibility breaking changes that necessitate the new major release, version 2.0. Substantial changes include refactoring the grid, improving the component standard interface, dropping Python 2 support, and creating 31 new components—for a total of 58 components in the Landlab package. We describe reasons why many changes were made in order to provide insight to designers of future packages. We conclude by discussing lessons about the dynamics of scientific software development gained from the experience of using, developing, maintaining, and teaching with Landlab.

# 1 Introduction

Landlab is a Python package to support the creation of numerical models in Earth surface dynamics. Numerical models support researchers to simulate past, present, and future dynamics of a system. This enables conceptual model validation, testing of alternative hypotheses, and prediction under uncertainty. Numerical modeling is especially important for Earth surface dynamics because of the timescale mismatch between human observation and system evolution. Landlab is an Open Source Python-language package that provides the common elements of infrastructure needed to support the creation of new models.

These include a model domain representation (the *model grid*), physical process *components*, and utilities that support use and extension of the package. Landlab's modular design lowers the barriers of entry to computational research, reduces researcher time, and supports publication of reproducible scientific research products (e.g., Bandaragoda et al., 2019). Development and maintenance of Landlab follows modern software development standards such as version control, integrated testing and documentation, continuous integration, and multi-platform binary distribution (e.g., Adorf et al., 2019; Hwang et al., 2017;

Mandli et al., 2016; Poisot, 2015; Taschuk and Wilson, 2017; Wilson et al., 2014). Our open source development and use of semantic versioning (SemVer 2.0.0, https://semver.org) provides a necessary but not sufficient tool for reproducible research in Earth surface dynamics (e.g., Chen et al., 2018).

   Landlab was designed as a key element in the Community Surface Dynamics Modeling System (CSDMS) suite of tools (Peckham et al., 2013). Initial development of Landlab began in 2012 and culminated in a version 1.0 release (referred to as

v1.0) described by Hobley et al. (2017). Figure 1 provides examples of the breadth of modeling efforts implemented with Landlab.

   Subsequent to the release of v1.0, the core development team and many community members have contributed additional features and bug fixes to the software. Based on experience using and developing with Landlab, the development team identified changes to Landlab that were not backwards compatible, indicating a major release was necessary to convey to Landlab users

to expect substantial changes. This motivated the creation of Landlab v2.0, the focus of this contribution. A new major version was needed to support (a) backward-compatibility breaking changes associated with refactoring core data structures, and (b) removal of Python<3 support.

   The scope of this contribution is to review the core concepts that underpin Landlab's design (Section 3), describe the changes and new features added since v1.0 (Section 4), discuss citation of software (Section 5), and document lessons we have learned

about community software development from developing and maintaining Landlab (Section 6). Before concluding we provide recommendations for those interested in being involved with Landlab (Section 7). We note that while much of the contribution discusses general issues of scientific software development, Section 4 is inherently specific to Landlab and describes technical details of changes between v1.0 and v2.0. For a comprehensive description of the design and theory behind Landlab v1.0 the reader is referred to Hobley et al. (2017). Additionally, we will not present detailed description of the use of the software,

discuss numerical methods, or review the literature that supports each process implemented in Landlab. In general, methods and supporting literature can be found in key publications introducing each component (see Section 5), and guidance on software usage can be found on the Landlab website.

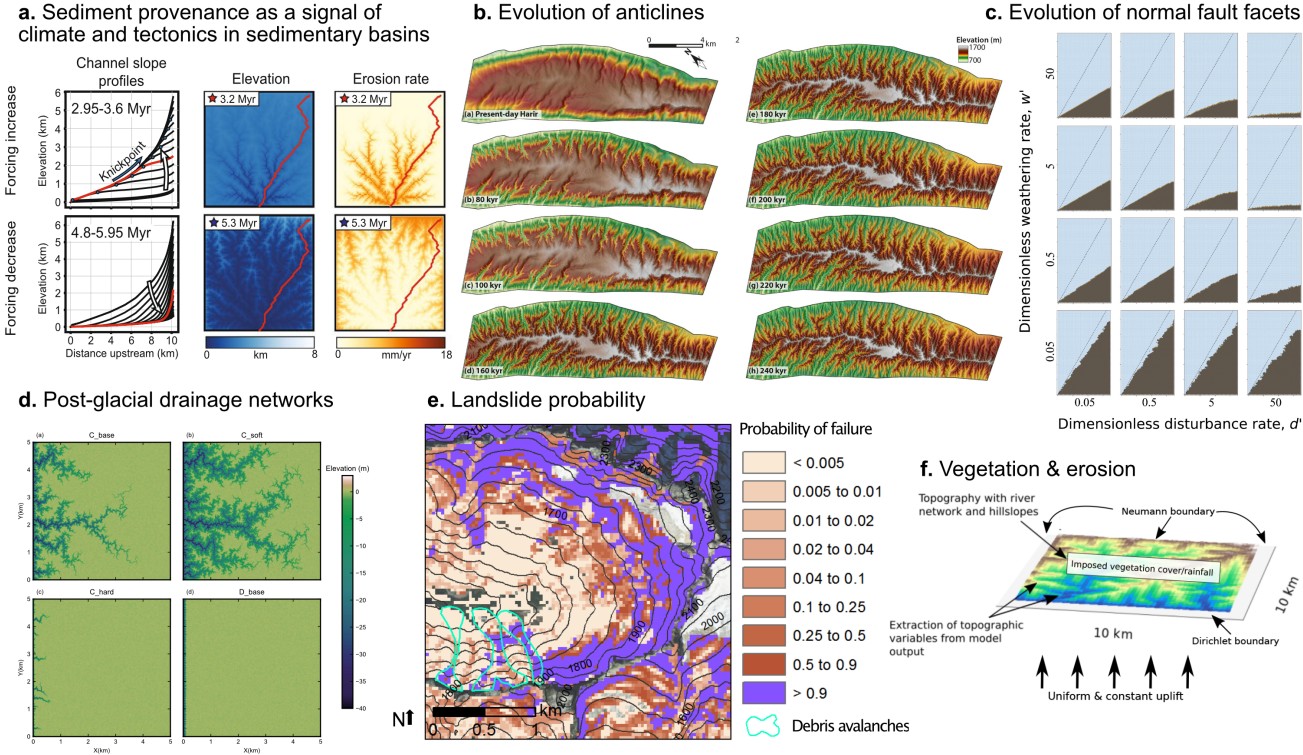

**Figure 1.** Examples of modeling applications implemented with Landlab span a wide range of timescales and topics. A recent selection of examples intended to highlight diversity of applications includes the following examples: (a) sediment provenance studies (Sharman et al., 2019, reproduction modified from their Figure 2), (b) landscape evolution of anticlines (Zebari et al., 2019, reproduction of their Figure 8), (c) cellular automaton simulation of normal fault facets (Tucker et al., 2020, reproduction of their Figure 9), (d) the evolution of post glacial drainage networks (Lai and Anders, 2018, reproduction of their Figure 4), (e) estimates of landslide probability (Strauch et al., 2018, reproduction modified from their Figure 9), (f) and coevolution of vegetation and erosion (Schmid et al., 2018, reproduction of their Figure 3). All sub-panels except (d) covered by CC BY license. Permission to reproduce (d) was obtained through Copyright Clearance Center RightsLink.

Detailed documentation for Landlab is available on the Landlab ReadTheDocs page https://landlab.readthedocs.io. Code availability is described at the end of the contribution. A PDF of the documentation and the source code for v2.0 are archived as the supplemental information to this publication.

## 2 The Three Landlab Audiences

The design of the Landlab package, its development practices, and the changes made in v2.0 are best understood in light of the three audiences who interact with the package. Unlike software that is developed by dedicated software engineers who may not

use the software themselves, Landlab developers also use the software for their research and teaching. Thus, the first audience is *user-developers*, people who extend, modify, or otherwise contribute to the source code in order to accomplish their goals. Notably, most Landlab user-developers have little to no background in software engineering. The second audience is *users*: people who use Landlab to write their own programs, but do not modify or contribute to Landlab's source code. Among this group, it is natural for some to transition to becoming user-developers, who contribute new components or utilities to the main Landlab code base. The final audience is *teachers-students*, people who use Landlab in an instructional classroom setting as part of a course.

In creating the source code, writing the documentation, determining the development practices, and maintaining the package, the needs, abilities, and time constraints of all three audiences must be balanced. This is particularly important for packages like Landlab with a small active developer community (n<20) and a research-scale user community (e.g., tens to hundreds of researchers and perhaps a few thousand students over the lifetime of the software, rather than millions of users). Our approach is to adopt many of the key design principles underlying modern academic software design best-practice (e.g., Wilson et al., 2017; The Turing Way Community et al., 2019). These include an extensive automatic test suite, integrated documentation, version control, continuous integration, lint checking, and releasing binary packages for users. These design choices were made to ensure that Landlab is sustainable into the future to support the user-developer-learner communities (see Hobley et al., 2017). Community contributors play an important role in developing community open source software. Two of their most important roles are improving and refining documentation when it is unclear, and identifying software bugs. Because Landlab currently has a relatively small user base with limited experience contributing to documentation, it takes longer (months to years) for documentation to be refined by users compared to software with more users (days to months). The relatively long "refinement residence time" means that a commitment to high quality tests is critically important (see Section 6.1).

## 3    Landlab Core Concepts

A core design principle behind the Landlab package is *modularity*. Separating the elements of a numerical model into reusable parts decreases the human-time associated with creating a new model or extending a current one. The design of Landlab is discussed extensively in Hobley et al. (2017). Here we briefly summarize the key points to provide context to the changes and new features that are discussed further in Section 4.

The modular design of Landlab comprises the following categories of software infrastructure:

1. *Model Grids*, data structures implemented as Python classes that represent the computational domain, connectivity between parts of the domain, and provide a centralized location to store state variables;

2. *Utilities*, functions that provide solutions to common problems (e.g., numerical functions for gradients, mapping, and flux divergence; basic plotting; watershed delineation; and file input/output).

3. *Components*, representation of core surface processes (e.g, stream power, flow accumulation, precipitation) as a Python class with a common interface.

The grid represents a 2D domain as a *dual-graph*. Each graph in the dual graph is a set of points, connected by lines, and outlining polygons. The two graphs are offset from one another such that the points of one graph are located inside of the polygons of the other graph. Each graph is a *planar graph*, meaning that the lines connecting points do not cross. In Landlab, we refer to the first graph as composed of *nodes* connected by *links* which outline *patches*. *Corners* are located inside patches and are connected by *faces*, which outline *cells*. With this framework, data identified at a given point in space has both a connectivity to other points defined by its lines, and a uniquely associated spatial area and set of bounding edges drawn the from enclosing polygon in the other graph (Figure 2).

There are four aspects of the grid that are worth highlighting. First is that the Landlab model grids provide information about the connectivity and adjacency of all grid *elements* (nodes, links, patches, corners, faces, and cells). Second, the model grids use a consistent framework for the numbering of grid elements and identifying a *direction* for each link and face (note that this is a topologic direction based on the orientation of the link in $x$-$y$ space, not a flow direction). This permits consistent application of numerical methods based on grid element ID that may be transferred to grids of different shapes and sizes.

Third, Landlab supports regular and irregular model grids through the same interface. The Landlab model grid library includes data structures for networks, regular rasters, general irregular meshes (Voronoi cells with Delaunay triangulated nodes), regular hexagons, and radially symmetric irregular meshes. Landlab v2.0 assumes all links and faces are straight. The model grids were designed to accommodate extension to more exotic 2D geometries.

Finally, the model grid may be used to store data *fields* at any grid element. Fields represent state variables and are useful when multiple components use or modify the state variables. When a field is stored on the grid, Landlab enforces characteristics such as the number of elements, and provides the ability to use adjacency information associated with the grid.

The Landlab model grids keep track of boundary conditions using arrays of integers with flags indicating characteristics such as fixed-value, fixed-gradient, or closed to flux (`grid.status_at_X` where `X` is the name of the grid element). Note that we will use `preformatted` style text to indicate Landlab syntax. Thus far, most applications with Landlab use nodes and links as the primary grid elements. Thus, sets of standard boundary condition flags are presently only implemented for these two types of grid elements.

Utilities fall into two subcategories: general numerical utilities, and application-focused utilities. In the first category are general functions to calculate quantities such as gradients or flux divergence, and map values from one grid element to another. Development has created numerical utilities focused on finite-difference/volume numerical solutions to differential equations and cellular automaton applications. The focus on finite-difference and finite-volume utilities, however, reflects the interests of developers rather than the potential characteristics of the package. In the second category are application-focused utilities. These utilities are typically developed for use in a particular component, but have grown to have broader use within the package. For example the watershed utility submodule was developed to support the `SpeciesEvolver`, but provides the broadly applicable ability to label watersheds.

Components are Python objects with a standard interface that implement a single Earth surface process, set of equations, or analysis compatible with the component interface (e.g. calculation of drainage density). All components require a Landlab model grid to instantiate, and have a built-in function that advances the component forward in time or updates it based on the

current values stored as fields. Components can be coupled by accessing and modifying the same fields stored on the model grid elements.

## 4   Changes and New Features Added Since Landlab v1.0

Landlab v2.0 contains many changes to the core source code that add new features. We have compiled tables describing the pre-existing, refactored, and new core capabilities of the Landlab package. These include data structures (Table 1), utilities (Table 2), new components (Table 3), and pre-existing or refactored components (Table 4). We list core package, development environment, testing, tutorial, and documentation dependencies in Table 5.

This section focuses on the technical details of what has changed between Landlab v1.0 and v2.0. One might glean comparable information from reading the software repositories' change logs. Inclusion of the technical details here is intended to summarize key changes. In addition, where relevant, we describe why changes or improvements were made. This explanation is intended both for current and future users of Landlab as well as for those interested in scientific software development generally.

Changes that broke backward compatibility were required to incorporate some of the new features in Landlab v2.0. This necessitated a new major version. These changes included: (i) binding of the boundary condition flags to model grids (Section 4.1.3), (ii) a revision to the Component standard interface (Section 4.2), (iii) deprecation and removal of certain components and utilities (Section 4.3), (iv) dropping Python 2 support (following sunsetting of this version at the end of 2019 by the Python Software Foundation). Additionally, we completely revised the documentation structure (Section 4.4). Landlab v2.0 is designed to work with a number of other Python tools for numerical modelling. They are summarized in Section 4.5.

### 4.1   Improvements to the Landlab Model Grids

Here we highlight three improvements to the Landlab model grid in v2.0.

#### 4.1.1   Grids Inherit from Graphs

Each Landlab model grid combines a dual-graph topology with the ability to store fields at grid elements and keep track of boundary conditions. While the concept of a dual-graph is not new in Landlab v2.0, the package architecture has been revised to create a set of graph classes from which the Landlab model grids inherit (Table 1).

The Landlab graphs describe the topology and connectivity of a dual graph of nodes-links-patches/corners-faces-cells, and specify the $x$ and $y$ coordinates of the nodes and corners. The package contains support for 1D and 2D graphs, and for graphs not yet used in Landlab grids (e.g., `DualStructuredQuadGraph`). It was designed to be re-usable by projects external to Landlab. While the graph capabilities do not yet support 3D graphs, the package was designed with extension to 3D in mind.

Building the model grids to inherit from the graph data structure results in all model grids containing a complete set of topology-derived attributes (e.g., `grid.links_at_node`) and attribute naming consistency between model grids. In addi-

**Table 1.** Major Data Structures in Landlab v2.0

| Name | Summary | New/Refactored? |
|------|---------|-----------------|
| **Graphs** | | |
| `NetworkGraph` | Graph with only nodes and links. | New |
| `DualVoronoiGraph` | Unstructured dual-graph of node-link-patch Delaunay triangles and corner-face-cell Voronoi polygons. | New |
| `DualHexGraph` | Dual-graph of node-link-patch triangles and corner-face-cell regular hexagons. | New |
| `DualRadialGraph` | Dual-graph with radially symmetric nodes. | New |
| `DualStructuredQuadGraph` | Dual-graph of structured quadrilaterals. Link and face lengths vary, and orthogonality of links and faces is not required. This graph does not yet support a grid. | New |
| `DualRectilinearGraph` | Dual-graph of quadrilaterals. Link and face lengths may be variable but angles are orthogonal. This graph does not yet support a grid. | New |
| `DualUniformRectilinearGraph` | Dual graph of constant-sized rectangles. $x$ and $y$ link and face lengths may be different, but are constant across the grid and are orthogonal. | New |
| **Model Grids** | | |
| `NetworkModelGrid` | Model grid that inherits from the `NetworkGraph` | New |
| `VoronoiDelaunayModelGrid` | Model grid that inherits from the `DualVoronoiGraph` | Refactored |
| `HexModelGrid` | Model grid that inherits from the `DualHexGraph` | Refactored |
| `RadialModelGrid` | Model grid that inherits from the `DualRadialGraph` | Refactored |
| `RasterModelGrid` | Model grid that inherits from the `DualUniformRectilinearGraph` | Refactored |
| **Other data structures** | | |
| `EventLayers` | Data structure that keeps track of a timeseries of thicknesses and a generic set of properties at all of one grid element (e.g., cells). In `EventLayers` every time point is recorded, such that erosion of layers retains a series of zero thickness. `EventLayers` is more appropriate if a user is interested in chronostratigraphy. | New |
| `MaterialLayers` | Same as `EventLayers` except that when erosion occurs, no layer is recorded, and when equivalent material is deposited, layers can be joined. | New |
| `DataRecord` | Data structure to store a generic set of variables in time and/or on grid elements. | New |

tion, all of the topology-derived attributes are only created when needed (just-in-time memory allocation) and are cached. This was inconsistently implemented in v1.0 and provides an improvement for memory management and speed.

The graph and model grid data structures are all built on the `xarray` Python package's `Dataset` (Hoyer and Hamman, 2016). Using `xarray.Dataset` provides a number of advantages including improved input and output to the NetCDF

**Table 2.** Major New Utilities in Landlab v2.0

| Submodule | Summary |
|---|---|
| `landlab.utils.distance_to_divide` | Calculate distance between nodes and watershed divides. |
| `landlab.utils.flow__distance` | Calculate distance between nodes and watershed outlets. |
| `landlab.utils.watershed` | Identify and label nodes that belong to individual watershed. |
| `landlab.values` | Create generic, reproducible, synthetic fields based on Python dictionaries or yaml input files. |

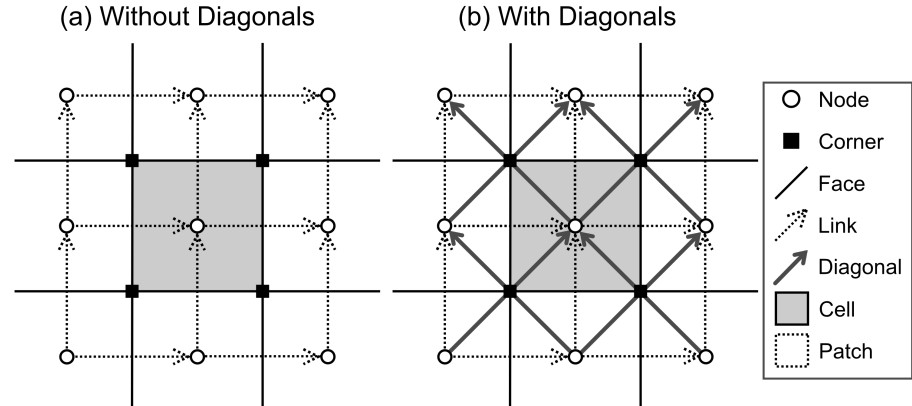

**Figure 2.** Grid elements of `RasterModelGrid` without (a) and with (b) diagonals.

format, use of `xarray`'s optimized data structures, and the possibility to take advantage of `xarray`-compatible parallelization related tools (e.g., `dask`, Dask Development Team, 2016; Rocklin, 2015) without breaking backwards compatibility.

#### 4.1.2   Improved Treatment of Diagonals

The `RasterModelGrid` can optionally contain an additional grid element called a *diagonal*, which connects nodes but also crosses corners (Figure 2). Including this grid element violates the assumption of a plane graph because the diagonal elements cross one another. However use of diagonal elements has a long history in Earth surface dynamics modelling; in order to support historical algorithms (e.g., D8 flow routing, O'Callaghan and Mark, 1984), Landlab's `RasterModelGrid` contains support for diagonals. This permits studies that cross-compare implementations with and without diagonals (e.g., Shelef and Hilley, 2013),.

Landlab v1.0 had a partial implementation of diagonals in which there was no consistent way to refer to the diagonals or the group of linear elements composed of both links and diagonals. In addition, we had an incomplete set of adjacency structures describing diagonals, and we had no mechanism to store values at diagonals on fields. We now consistently call the set of links and diagonals *d8s*, and have implemented adjacency structures and some numerical functions for diagonals and d8s that mirror those for links. Landlab assigns a unique ID to each grid element (see Hobley et al., 2017, their Figure 4). For example, the nodes are identified with ID numbers from zero to number of nodes minus one, and links are identified with numbers from zero to number of links minus one. The unique IDs assigned to the d8s refer first to the links and then to the diagonals (in this contribution we will use "d8" to refer to the grid element and "D8" to refer to the flow routing approach).

### 4.1.3 Boundary Condition Flags

Landlab v2.0 provides boundary condition status arrays for nodes, links, corners, faces, and, if applicable, diagonals and d8s. Because cells and patches are uniquely associated with their own nodes and corners, we do not supply specific status arrays for those elements. Boundary condition status is indicated by a set of flags that indicate the status (Table 6 indicates flag names, see Hobley et al., 2017, their Section 3.1.4 for discussion of boundary conditions). Landlab does not enforce whether a component honors boundary condition flags—the status arrays and flags are provided simply as a convenience to developers. As in v1.0, we enforce internal consistency of boundary conditions across connected grid element types. For example, an update to boundary status at a node will automatically propagate into the connecting links as appropriate, and vice versa.

Prior to v2.0, the flags used to indicate node and link status were not formally attached to the model grids and instead existed as separate variables provided by the package. In v2.0 we made boundary condition flags attributes of the grid so that these flags are inseparable from the grids that use them. We also modified the names for clarity (Table 6).

### 4.2 Updates to the Component Standard Interface

Scientific software and data are much easier to work with when they follows standards. Software tools in particular become much more accessible when they provide a *standard interface*: a common set of functions that look and act in a similar way across many different elements of the software. Landlab's components use a lightweight interface that is inspired by the CSDMS Basic Model Interface (BMI) (Peckham et al., 2013; Hutton and Piper, 2020a), but which takes advantage of object-oriented features of the Python language, allowing it to be more compact. (Landlab also includes built-in functionality that converts any Landlab component into a BMI component, for use in frameworks like CSDMS' Python Modeling Tool (Hutton and Piper, 2020b)). In addition to its interface, each Landlab component also encodes metadata in a standardized format; these metadata include, for example, information about the component's input and output fields.

We made changes to the expectations of component interface, metadata, and code standards based on our experience developing components, supporting community members, and using components in science applications. The enhanced interface standard is designed to improve usability and documentation, and to make clearer expectations for contributed components. We have implemented automated tests that ensure existing and contributed components meet this interface standard.

### 4.2.1  Changes to the Component `__init__` Method

The design of many numerical model programs follows the "initialize-run-finalize" pattern (e.g., Peckham et al., 2013; Hutton and Piper, 2020a). In the Basic Model Interface, the initialization step is handled by the standard `initialize()` function, and stepwise updating is handled by the `update()` function. For Landlab components, which are implemented as Python objects, the class `__init__` method implements initialization, and stepwise updating is normally handled by a method called either `run_one_step` or `update`. Hobley et al. (2017, their section 3.3.1) defined the interface for Landlab components with the function signature for instantiation (`Component.__init__`) and advancing forward (`Component.run_one_step`). The v1.0 component instantiation interface defined with the function definition of: `__init__(self, grid, arg1, arg2..., kwd1=a, kwd2=b, kwd3=c, ..., **kwds)`. Here `arg1` represents a generic argument and `kwd1=a` represents a generic keyword argument. The `**kwds` was included so that a user could make a single dictionary (or yaml file) containing all of the keyword arguments for all components used in a model, and pass the same dictionary to all components. However, an undesirable side effect of this design was that a slight misspelling of a keyword argument would result in use of the default value with no error raised. To remedy this flaw we revised the instantiation standard to remove the `**kwds`; that is, a user may now only supply the component with input parameters that are explicitly declared in its signature.

In addition we expanded the requirements for component instantiation. These requirements help promote standardization among Landlab components. One new requirement is that components must inherit from the `Component` base class and call the instantiation method of the base class (`super`) as part of their instantiation. This ensures that all components take full advantage of the base class functionality and internal checking; for example, the base class will automatically make sure that all of the output fields listed in the header metadata are created, so the component author only needs to ensure that the metadata are present. A related requirement is that by the end of instantiation, all output fields made by the component must exist and have the data type specified by the component metadata. This provides for other components that may check for these fields as input. Finally, a component must raise a sensible error when bad values are provided (for example if an unsupported grid type or unused keyword argument is provided).

### 4.2.2  Changes to the Component Run Method

The v1.0 component interface defined a run method with a function signature `run_one_step(dt, *args, **kwds)` where `dt` represents the duration of time the model runs forward, `*args` represents a generic list of arguments, and `**kwds` represents a generic set of keyword arguments. In practice, we found that many Landlab components were not able to follow this interface standard because it was not flexible enough. For example, some components do not require a `dt` and thus did not take `dt`. We also found the presence of `*args` and `**kwds` in the `run_one_step` problematic because it complicated wrapping components with a Basic Model Interface (BMI, Peckham et al., 2013; Hutton and Piper, 2020a) for use with the Python Modeling Tool (PyMT, Hutton and Piper, 2020b).

The revised interface balances standardization and flexibility. Components are no longer required to provide a method with the name `run_one_step`, but if they do not, then an alternative update/execution function must be provided and its usage

clearly documented in the component's header docstring. The new expectation is that if `run_one_step` is used it will either take a time duration or nothing. Thus components with a `run_one_step` method can be easily incorporated into PyMT. Pre-existing components that took arguments or keyword arguments in their `run_one_step` method have been refactored to either provide those values at instantiation, or to use *properties, getters,* and *setters*. The terms getter and setter come from object-oriented programming, and they refer to small functions that retrieve the value of (get) or assign a value to (set) a particular variable. Although it might seem odd to create functions to handle such seemingly trivial tasks, the practice has the advantage of enabling defensive programming (e.g., a setter can check for the right data type), allowing a program to create a particular variable only when it is requested (which can save memory), and supporting built-in documentation (in the form of function documentation) for each variable. In Landlab (as in Python practice generally) getters and setters are implemented using the Python `@property` decorator. Those variables that use getters and setters are considered to be public, meaning that programmers using the component can easily inspect and, if desired, change their values. Other variables are considered private: used only by the component internally, and not to be modified (to indicate this, the names of private variables are preceded by an underscore character).

### 4.2.3 New Component Metadata Standard

For both data and software, standardized metadata promote efficiency, interoperability, and reuse. To that end, each Landlab component includes a set of metadata in the header of the class that defines the component. Our experience with component metadata in Landlab led us to revise its design for version 2.0.

The metadata section of a Landlab component describes its input fields, output fields, field units, the type of grid element associated with each field, and a long-format description of the field. Metadata are now organized into a single Python dictionary, which has a key-value pair for each field used by the component. The new data structure makes it easier to test for completeness and consistency across components. Each key is a string indicating the field name. The associated value is itself a dictionary that has a standard, required set of keys (Table 7).

### 4.2.4 Additional Component Content Requirements and Recommendations

Here we highlight the few remaining component requirements and recommendations. The use of *must* indicates a requirement while the use of *may* or *should* indicates a recommendation.

- All public attributes must be documented *properties* of the Component class, that is, they have the `@property` standard Python decorator. This ensures that other users are able to identify what each public attribute is, and prevents variable modification unless the developer explicitly permits it. This change has little impact on developers time because a developer may elect to use only private attributes.

- If a developer envisions that a component's public attribute may be modified, they must create a `setter` for it. This provides a place for a component author to write checks that ensure a user cannot incorrectly assign invalid component attributes.

– Field names shared between multiple components must use a consistent definition and dimensions. Some components require parameters and fields to use a particular set of units while others are unit agnostic. This is flagged in the component attribute `Component.unit_agnostic`. It is up to the user to ensure that an application uses consistent units across all fields, components, and input parameters.

– Arguments and keyword arguments should start with lower case letters.

– The grid should be the only argument to the component `__init__`. All other inputs are provided as keyword arguments.

– Keyword arguments should have reasonable default values so that all keywords are truly optional.

– The component's main method (either `run_one_step` or a custom-designed update/execution function) should return either nothing, the grid, or a single calculated value.

**4.3 Removed or Modified Components and Utilities**

Several obsolete components and utilities have been removed from Landlab v2.0. Other components were substantially modified. Here we describe these changes.

– The `FlowRouter` component, which did D8 and D4/Steepest Descent flow routing and accumulation, was removed and replaced with the `FlowAccumulator` and a family of `FlowDirector` components. This change provides greater
flexibility in options for flow-routing algorithms (e.g., multiple flow directions, $D_\infty$).

– The routing-based surface-water erosion components (such as `StreamPowerEroder`) now use a single consistent method for handling the input runoff rate. The keyword argument `runoff_rate` to the `FlowAccumulator` can now specify a float, array, or field name indicating the runoff rate. This is then accumulated to create the field `surface__water_discharge` which can be used by components that model surface-water erosion.

– The `ModelParameterDictionary` was removed because it represents an old-style input file that has been superceded by the yaml format.

– A new `ChannelProfiler` component replaces the previous channel-profiling submodule (`landlab.plot.channel_profile`).

– The `noclobber` keyword argument for field creation was changed to `clobber` because the original name required
double negatives and was not intuitive. `noclobber=False` is equivalent to `clobber=True`.

– The ability to pass an array of flooded nodes to the `run_one_step` method in surface-water erosion components was removed and replaced with a keyword argument to `__init__` called `erode_flooded_nodes`.

### 4.4 Reorganization of the Landlab Documentation

The Landlab online documentation is now consolidated onto a single sphinx-based platform (https://landlab.readthedocs.io/). Consolidating the documentation onto a single platform with a consistent interface reduces duplication of information, improves consistency, and permits comprehensive searches. The site's design is similar to that of widely used scientific Python packages and was modeled after that of `pandas` (McKinney, 2010). The revised documentation pages include installation instructions, a User Guide (including tutorials), a Guide for Developers, and an API Reference that contains formatted versions of inline documentation within the source code. The documentation source is written in ReStructuredText format, and the source files are provided as part of the Landlab package.

### 4.5 Packages Built to Work With Landlab

Landlab was designed as a generic, extensible modelling framework for Earth surface dynamics. Because Landlab exposes BMI (BMI, Hutton and Piper, 2020a), it is compatible with the PyMT package (Hutton and Piper, 2020b)—a Python package that supports running and coupling models that expose a BMI. PyMT provides access to a suite of models written in multiple languages (e.g., Python, fortran, c++) and a standard interface for initializing and running.

In addition, two packages have been built using Landlab to support applications in sensitivity analysis, calibration, validation, and multi-model comparison (see, Barnhart et al., 2020a, b, c, for example applications). First, `terrainbento` is a Python package for multi-model analysis that provides an extensible set of 27 Landlab-built models for long-term drainage basin and landform evolution, along with general classes for handling boundary conditions through an input-file format (Barnhart et al., 2019b). Second, `umami` is used to calculate model-data comparison metrics for observed and simulated topography (Barnhart et al., 2019a).

### 5 Citation of Landlab and Parts of Landlab

Citation of scientific software is an outstanding challenge (e.g., Niemeyer et al., 2016). Scientific software is cited less frequently than it is used (e.g., Pan et al., 2015). Indicating a recommended citation for use of Landlab is additionally challenging because, depending on the portion of Landlab used, the set of citations required may vary. We describe our recommendations for which citations to use, and present a tool within Landlab to improve citation discoverability.

Any time any part of Landlab is used, Hobley et al. (2017) should be cited; if the version used is $> 1.0$, then this contribution should be additionally cited. These two citations acknowledge the development of the Landlab package itself. We also recommend that authors state the specific version of Landlab used (the version can be found by evaluating `landlab.__version__`).

Each application of Landlab may use a different set of components, each with a different citation for the software itself and general set of theory references (Table 3 and 4). Additionally, some parts of Landlab may internally use others; thus a user may not easily be able to assess the entire set of elements of Landlab their application has used and what to cite for each part.

This challenge is not new. For example, it is faced by the `scipy` package, which addresses it by providing a core-package citation: Virtanen et al. (2019), and indicating that users should look to the Reference section of the documentation for additional citations. Similarly, the codes distributed through the Computational Infrastructure for Geodynamics (CIG) have a citation builder that distinguishes between citations specific to the software implementation, primary citations describing the code development and numerical methods, and secondary citations that pertain to parts of the code a user may or may not have used (Kellogg et al., 2018). This example from CIG highlights a further challenge: a component may have one or more citations for for each of the following categories: (i) the theory behind the implemented idea, (ii) a description of the software implementation itself, (iii) any specialized algorithms developed for the implementation, and (iv) the first reported use of the software in a publication.

Should one of these or all of these be the recommended and/or required citations for a given software component? We do not think it is our role to decide which citations, if any, a component author indicates as recommended or required. Additionally, it is not our place—as the software developers behind Landlab—to determine which citations best represent the theory behind an implementation. Instead we provide two places for a component author to indicate what they think the minimum required citations are: a component attribute called `Component.cite_as` which lists required citations for a given component, and a section in the component docstring that provides the broader reference context. These two categories are reflected by the two citation columns in Tables 3 and 4. Clearly, a component developer has the authority to decide exactly what to put in either of these locations.

To aid discoverability of citations, we have created the Landlab *citation registry*, a tool that compiles citation-related metadata for the specific set of Landlab components used in an application (Listing 5). The citation registry compiles citation information for all components currently instantiated in a Python session by automatically interrogating their `cite_as` properties.

## 6 Lessons on Geoscientific Software Development

In this section we highlight several lessons about software development we have learned in the processes of supporting and improving Landlab v1.0 to its current v2.0 state and working with the growing community of users.

We reflect on these lessons because the production of research software is itself research and there are many aspects of scientific software which are distinct from other software, notably (i) that the development lifecycle includes additional stages because the methods used to implement a piece of software may not exist at the outset of a project, (ii) requirements evolve because they are part of the research, and (iii) the state of the scientific field may be complex and evolving (e.g., Carver et al., 2016).

### 6.1 Value of Testing

The development of docstring and unit tests within Landlab was motivated by following software development best practices (e.g., Wilson et al., 2014, 2017). That is, our focus was on ensuring that the package behaves as described and, where an

**Listing 1.** Using the Landlab citation registry.

```python
import landlab

# Do your work, using the parts of Landlab you need.

# When you are done, write citations to a file.
w = landlab.registry.format_citations()
with open("citations.bib", "w") as f:
f.write(w)

# This will produce Bibtex-formatted citations for all
# Landlab components that you currently have
# instantiated.

# For example, the Bibtex contents below lists the
# first entry from a script that only imports Landlab
# (this contribution would also be listed).

# Citations
## landlab
@article{hobley2017creative,
AUTHOR = {
Hobley, D. E. J. and Adams, J. M. and Nudurupati,
S. S. and Hutton, E. W. H. and Gasparini, N. M. and
Istanbulluoglu, E. and Tucker, G. E.
},
TITLE = {
Creative computing with Landlab: an open-source
toolkit for building, coupling, and exploring two-
dimensional numerical models of Earth-surface dynamics
},
JOURNAL = {Earth Surface Dynamics},
VOLUME = {5},
YEAR = {2017},
NUMBER = {1},
PAGES = {21--46},
URL = {https://www.earth-surf-dynam.net/5/21/2017/},
DOI = {10.5194/esurf-5-21-2017}
}
```

analytical solution exists, that Landlab correctly solves it. While using a testing suite is standard in many software development contexts, it is relatively uncommon in scientific software development (e.g., Prabhu et al., 2011). Tests do not ensure that elements of the Landlab software represent the truth, or guarantee that a model is appropriate for a specific application; in other words, Landlab cannot and does not attempt to *validate* (sensu Schlesinger et al., 1979) the assumptions of its components. Instead, the tests *verify* (Schlesinger et al., 1979) that the software is behaving as expected and that numerical methods are

solving stated equations reliably. Through coupled use of an automatic testing suite and continuous integration we ensure that changes to the code base do not break existing tests.

The process of developing Landlab, working with its user community, and revising it to v2.0 illustrated another, obvious in retrospect, benefit of the tests: developing a set of tests for the package interface and numerical behavior *make it possible to refactor*. Without these tests, it would have been much more difficult to implement beneficial revisions (such as refactoring the

360 model grid to derive from the graph-based class).

Writing effective unit tests that ensure Landlab components reliably solve their equations under a variety of initial and boundary conditions is not a trivial task. When a set of equations that a component solves have an analytical solution then the numerics of a component can be verified based on the ability to reproduce such a relationship (e.g., stream power erosion produces a known slope-area relationship Willgoose et al., 1991). When such analytical predictions do not exist—as is often

the case—a more detailed analysis of the equations must be performed in order to create a full verification test. Even in the absence of such analytical solutions, however, many existing Components have made headway during development simply by testing for mass balance and timestep consistency, and the value of such simplifications should not be ignored.

In contrast, it is much easier to design and implement tests for the Landlab *interface* (e.g., when a invalid value is passed to a component, is the correct type of error raised). In general, designing a thorough set of tests is a learned skill that requires

thinking through many edge cases of model behavior.

### 6.2 Collaborative Development of Research Software Requires Many Skills

Scientific software development requires distinct skills. Based on working with community user-developers and onboarding new members of the core development team, we describe the set of skills that are needed to interact with a project like Landlab as a user-developer. Our intention here is to document a concrete example so that efforts to create scientific software

development curricula can be based on use-cases. In the case of Landlab, the skills required to contribute to the project include:

1. Python programming, including functions, classes, and basic package organization.

2. Fundamental elements of version control using git (branching, commits).

3. GitHub for collaboration (issues trackers, merging, pull requests, managing forks, code reviews).

4. Package dependency management (currently implemented with conda environments).

5. Conceptual design and practical implementation of unit tests.

6. ReStructured text syntax for creating documentation.

In addition, there are a number of skills that not all user-developers need, but are necessary to have within the project team in order to maintain continuous integration, documentation, building binaries, and distributing (e.g., `sphinx`, configuring and debugging continuous integration platforms).

The importance of these skills is highlighted in the context of *technical debt*, or the cost of implementing a fast and easy solution now, as opposed to a better approach that may take longer. For example, we have found that it is much easier to create content than to make it accessible (this observation motivated the restructuring of the documentation described in Section 4.4). It is also easier to write code than to write thorough and effective tests for it, yet omitting tests greatly increases the risk of serious bugs, which can invalidate the research that the software is meant to facilitate.

**6.3   Balancing the Burden on Developers and Users**

Open-source software (scientific or otherwise) commonly has many more users than developers or user-developers (e.g., `numpy`). Under those circumstances, moderate investments in developer time are justified to make use faster or more intuitive for users. However, Landlab is a case with slightly different dynamics, which are worth reflecting on. Landlab is an example of a niche scientific software package with a relatively small development community. Here we reflect on some of the

development dynamics of this type of scientific software and the relative burdens for use on developers and users.

Our goal is to create an extensible software package that solves a variety of Earth surface dynamics problems and is accessible to undergraduates and active researchers, *and* to support community members in contributing to the code (transitioning from users to user-developers). Effectively serving the community requires a balance between minimizing technical debt (by enforcing standards within the code base), while also making development and contribution accessible to inexperienced but

motivated community members.

One aspect of our approach, inspired by experience working with community members, is to be flexible with the software engineering and interface standards. This includes relaxing standards when necessary. For example, while a strict interface standard for components would likely reduce technical debt, our experience is that such rigidity would raise a substantial barrier to community contribution. This means that we need to strike a balance in our design principles between standardization and

flexibility (e.g., relaxing the standard for the `run_one_step` method described in Section 4.2).

Second, we embrace the idea that good is better than not at all. That is, some tests are better than none, meaningful tests are better than non-meaningful ones, and barebones documentation is better than none. We find that documentation improves the most when users try to use it, find that it is insufficient or unclear, and interact with developers through the online and open GitHub Issues forum. Users and developers then together revise the text. Because the development team is small and supported

primarily by grants, we rely on users to indicate where improvements must be made.

## 7    How Do I Get Started?

We highly encourage all contributions to Landlab. The package is designed as an extensible piece of community software and we look forward to it growing to meet community needs. Common ways that an interested individual might get started include: identifying or making improvements to the documentation and example notebooks, finding and fixing bugs, and describing and creating desired features—such as new components. For information about how to get started, visit the website at https://landlab.readthedocs.io/.

## 8    Conclusions

Landlab v2.0 provides the community with a robust and extensible package for modelling Earth surface dynamics. It is distributed as source code and as pre-packaged binaries for Linux, MacOS, and Windows. An extensive set of unit tests ensure reliability of the code base. This version provides substantial improvements over the v1.0 including (i) a revised set of model grid classes, (ii) updates to the component interface, (iii) 31 new components, (iv) expanded and consolidated documentation, and (v) a tool for identifying appropriate citations. The backward-compatibility breaking changes made in Landlab v2.0 reflect changes necessary based on use and development of the package. The modular design of Landlab means that developers only need to create the new piece they need, and researchers can mix and match components to create a desired model. As a tested, version-controlled, and documented software package, Landlab reduces barriers to computational modelling and supports reproducible research.

*Code availability.*    The Landlab source code is hosted on GitHub at https://github.com/landlab/landlab. Our documentation can be found at https://landlab.readthedocs.io/. Prepackaged binaries are distributed through PyPI (https://pypi.org/project/landlab/) and conda-forge (https://anaconda.org/conda-forge/landlab). The v2.0 version of the software and a PDF of the documentation are provided as a supplement to this contribution and are archived with Zenodo (Hutton et al., 2020).

*Author contributions.*    KRB and EWHH led the design and v2.0 refactoring of the Landlab package with input from all co-authors. KRB wrote the original draft of the manuscript, with input from all co-authors. All authors edited the manuscript. KRB, EWHH, GET, NMG, DEJH, NJL, MM, SSN, and JMA contributed to the Landlab code base. All authors designed and taught short courses which provided usability testing and resulted in critical improvements to package architecture and documentation. CB expanded accessibility of Landlab using advanced cyberinfrastructure by leading integration of Landlab with the Hydroshare platform. GET, NMG, EI, and EWHH conceptualized Landlab and created its prototype. GET, NMG, EI, and DEJH acquired the core funding to support Landlab, with additional funding acquired by KRB, CB, and NJL.

*Competing interests.*    The authors declare no competing interests.

*Acknowledgements.* Landlab was supported by the following US National Science Foundation awards: 1147454 (GET), 1450409 (GET),
1147519 (NMG), 1450338 (NMG), 1148305 (EI), 1450412 (EI), 1246761 (through an NCED2 postdoctoral fellowship to DEJH), 1725774
(an EAR postdoctoral fellowship to KRB), and 1902600 (CB). Landlab is additionally supported by the Community Surface Dynamics
Modeling System (NSF Award Numbers 1226297 and 1831623). DEJH acknowledges the support from a Marie Curie/Ser Cymru II Cofund
Research Fellowship 663830-CU-035, and from a Software Sustainability Institute Fellowship. NJL and NMG acknowledge the support of a
Tulane University Oliver Fund Scholar Award. We thank Tristan Salles and Wolfgang Schwanghart for thoughtful reviews, and Simon Mudd
for serving as handling editor.

We acknowledge support from Tony Castronova and the HydroShare platform at Consortium of Universities for the Advancement of
Hydrologic Science, Inc. (CUAHSI). CUAHSI supports use of Landlab on the HydroShare Platform (NSF EAR 1338606). Landlab Group
members on HydroShare have freely shared research, data, training and teaching resources with Landlab and HydroShare communities.
Landlab relies on free open-source package builds from TravisCI and Appveyor for our Continuous Integration. Our documentation is hosted
for free by ReadTheDocs.

Landlab would not exist without decades of Open Source software development. In this spirit, we thank all community members who have
asked questions, made Issues, commented on documentation that didn't make sense, and contributed code to the package. Below we list the
results of our best efforts to compile all non-author community contributors to the Landlab package. The are (in alphabetical order): Guiseppe
Cippolla, Jon Czuba, Vanessa Gabel, Rachel Glade, Jenny Knuth, Abby Langston, David Litwin, Amanda Manaster, Allison Pfeiffer, Francis
Rengers, Charlie Shobe, and Rhonda Strauch.

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

**Table 3.** New components added since Landlab v1.0

| Component | Summary | Required Citation[1] | Additional References |
|---|---|---|---|
| ChannelProfiler | Extract channel networks | | |
| DepthDependentDiffuser | Linear, depth-dependent diffusion of topography | Barnhart et al. (2019b) | Johnstone and Hilley (2015) |
| DepthDependentTaylorDiffuser | Nonlinear, depth-dependent diffusion of topography | Barnhart et al. (2019b) | Johnstone and Hilley (2015); Ganti et al. (2012) |
| DischargeDiffuser | Diffuse sediment proportional to an implicit water discharge value | | |
| ErosionDeposition | Fluvial erosion and transport | Barnhart et al. (2019b) | Davy and Lague (2009) |
| ExponentialWeatherer | Calculate weathering rate based on exponential function of soil thickness | Barnhart et al. (2019b) | Ahnert (1976); Armstrong (1976) |
| Flexure1D | 1D lithospheric flexure under loading | | |
| FlowAccumulator | Calculate drainage area and discharge | | Braun and Willett (2013) |
| FlowDirectorD8 | Direct flow based on D8 scheme | | O'Callaghan and Mark (1984) |
| FlowDirectorDINF | Direct flow based on $D_\infty$ scheme | | Tarboton (1997) |
| FlowDirectorMFD | Direct flow to multiple downstream receivers | | Quinn et al. (1991); Freeman (1991) |
| FlowDirectorSteepest | Direct flow based on D4 scheme | | |
| GroundwaterDupuitPercolator | Model flow in a shallow unconfined aquifer using the Dupuit-Forcheimer approximation | Litwin et al. (2020) | Childs (1971); Marçais et al. (2017) |
| HackCalculator | Calculate Hack's law parameters for drainage basins | | |
| LakeMapperBarnes | Identify and route flow through lakes | Barnes et al. (2014) | |
| LandslideProbability | Simulate landslide probability of failure, mean relative wetness, and probability of saturation | Strauch et al. (2018) | |
| LateralEroder | Lateral erosion of fluvial channels | Langston and Tucker (2018) | |
| LithoLayers | Manage layered material with variable properties | Barnhart et al. (2018) | |
| Lithology | Manage material with spatially variable properties | Barnhart et al. (2018) | |
| LossyFlowAccumulator | Calculate drainage area and discharge, while permitting dynamic loss or gain of flow downstream | | Braun and Willett (2013) |
| NormalFault | Vertical uplift on a generic fault | | |
| PotentialityFlowRouter | Calculate a discharge field using a matrix solution | | |
| Profiler | Extract generic profiles across a Landlab field | | |
| SinkFillerBarnes | Fill depressions in a surface | Barnes et al. (2014) | |
| Space | Fluvial erosion by stream power with alluvium conservation and entrainment | Shobe et al. (2017) | |
| SpatialPrecipitationDistribution | Generate spatially resolved precipitation events | Singer et al. (2018) | |
| SpeciesEvolver | Evolve life in a landscape | Lyons et al. (2020) | Albert et al. (2016); Lyons et al. (2019) |
| StreamPowerSmoothThresholdEroder | Fluvial erosion with a smoothed-threshold version of stream power | Barnhart et al. (2019b) | Braun and Willett (2013) |
| TaylorNonLinearDiffuser | Nonlinear diffusion of topography | Barnhart et al. (2019b) | Ganti et al. (2012) |
| TransportLengthHillslopeDiffuser | Non-local hillslope diffusion | | Davy and Lague (2009); Carretier et al. (2016) |
| TrickleDownProfiler | Extract profiles downstream of arbitrary points | | |

1. In addition to Hobley et al. (2017) and this contribution

**Table 4.** Landlab components in v1.0 (after Hobley et al. (2017), their Table 5)

| Component | Summary | Required Citation[1] | Additional References |
|---|---|---|---|
| ChiFinder | Calculates the chi index along a channel network | | Perron and Royden (2012) |
| DepressionFinderAndRouter | A lake filler that can route flow across depressions | | Tucker et al. (2001b) |
| DepthSlopeProductErosion | Detachment limited fluvial erosion calculated using depth-slope product for shear stress | | |
| DetachmentLtdErosion | General implementation of detachment limited fluvial erosion | | Howard (1994) |
| DrainageDensity | Calculate drainage density | | Tucker et al. (2001a) |
| FastscapeEroder | Implements fluvial erosion according to stream power, using the Fastscape algorithms | | Braun and Willett (2013) |
| FireGenerator | Produces intervals between fire events, following a Weibull distribution | | Polakow and Dunne (1999) |
| Flexure | Simple lithospheric flexure under loading | Hutton and Syvitski (2008) | Lambeck (1988) |
| FractureGridGenerator | Generate fractures in a model grid | | |
| gFlex | A more complex flexure model, utilizing gFlex | Wickert (2016) | |
| KinwaveImplicitOverlandFlow | Locally implicit implementation of the two-dimensional kinematic wave model | | |
| KinwaveOverlandFlowModel | Simple implementation of the two-dimensional kinematic wave model | | |
| LinearDiffuser | Linear diffusion of topography | | Culling (1963) |
| OverlandFlow | An inertial approximation of the shallow water equations for overland flow applications | Adams et al. (2017) | de Almeida et al. (2012) |
| OverlandFlowBates | An inertial approximation of the shallow water equations for overland flow application | | Bates et al. (2010) |
| PerronNLDiffuse | Nonlinear hillslope diffusion | | Perron (2011) |
| PotentialEvapotranspiration | Calculate potential evapotranspiration across a surface | | ASCE (2005); Zhou et al. (2013) |
| PrecipitationDistribution | Generate a storm sequence of intervals and intensities | | Eagleson (1978) |
| Radiation | Calculate total incident shortwave solar radiation | | Bras (1990) |
| SedDepEroder | Sediment-flux-dependent shear stress based fluvial incision | | Hobley et al. (2011) |
| SinkFiller | An algorithm to fill depressions in a surface | | Tucker et al. (2001c) |
| SoilMoisture | Compute local inter-storm water balance and root-zone soil moisture saturation fraction | | Laio et al. (2001) |
| SoilInfiltrationGreenAmpt | Infiltrate surface water into a soil following the Green-Ampt method | Rengers et al. (2016) | Julien et al. (1995) |
| SteepnessFinder | Calculates steepness indices for a channel network | | Wobus et al. (2006) |
| StreamPowerEroder | Implements fluvial erosion according to stream power, using the Fastscape algorithms | | Braun and Willett (2013) |
| VegCA | Cellular automata algorithm to simulate spatial organization of plant functional types | | Zhou et al. (2013) |
| Vegetation | Calculate above-ground live and dead biomass, and leaf area index | | Zhou et al. (2013) |

1. In addition to Hobley et al. (2017) and this contribution.

**Table 5.** Dependencies and Citations

| Category | Name | Citation |
|---|---|---|
| Core Package | `bmipy` | Peckham et al. (2013); Hutton and Piper (2020a) |
| | `matplotlib` | Hunter (2007) |
| | `netcdf4` | Whitaker et al. (2019) |
| | `pyyaml` | |
| | `pyshp` | |
| | `scipy` | Virtanen et al. (2019) |
| | `statsmodels` | Seabold and Perktold (2010) |
| | `pandas` | McKinney (2010) |
| | `xarray` | Hoyer and Hamman (2016) |
| Testing | `coveralls` | |
| | `pytest` | Krekel et al. (2004) |
| | `pytest-cov` | |
| | `pyyaml` | |
| | `pytest-datadir` | |
| Tutorials | `dask` | Dask Development Team (2016); Rocklin (2015) |
| | `jupyter` | Pérez and Granger (2007); Kluyver et al. (2016) |
| | `holoviews` | |
| | `nbformat` | |
| Development | `black` | |
| | `flake8` | |
| | `isort` | |
| Documentation | `sphinx` | |
| | `sphinx_rtd_theme` | |
| | `pandoc` | |
| | `tornado` | |
| | `entrypoints` | |

**Table 6.** Boundary Condition Flag Changes

| Landlab v1.0 Name | Landlab v2.0 Name |
|---|---|
| `BAD_INDEX_VALUE` | `ModelGrid.BAD_INDEX` |
| `CORE_NODE` | `ModelGrid.BC_NODE_IS_CORE` |
| `FIXED_VALUE_BOUNDARY` | `ModelGrid.BC_NODE_IS_FIXED_VALUE` |
| `FIXED_GRADIENT_BOUNDARY` | `ModelGrid.BC_NODE_IS_FIXED_GRADIENT` |
| `LOOPED_BOUNDARY` | `ModelGrid.BC_NODE_IS_LOOPED` |
| `CLOSED_BOUNDARY` | `ModelGrid.BC_NODE_IS_CLOSED` |
| `ACTIVE_LINK` | `ModelGrid.BC_LINK_IS_ACTIVE` |
| `INACTIVE_LINK` | `ModelGrid.BC_LINK_IS_INACTIVE` |
| `FIXED_LINK` | `ModelGrid.BC_LINK_IS_FIXED` |

**Table 7.** Metadata for fields component fields

| Name | Description |
| --- | --- |
| "dtype" | The data type for the items in the field indicated as a Python data type (e.g., float, int). |
| "intent" | A string indicating the input/output intent of the field. Valid options are "in", "out", and "inout". |
| "optional" | Boolean indicating whether the field is an optional input or output. |
| "units" | String indicating the units of the field. Some components are unit agnostic, in which case these units can be interpreted as dimensions (see item below describing the attribute `Component.unit_agnostic`). |
| "mapping" | String indicating the type of grid element associated with the field (e.g., node, link). |
| "doc" | String describing the field. |