# Peer review of "Short communication: Landlab v2.0: A software package for Earth surface dynamics"

_Earth Surface Dynamics, 2020_

## Referee Comment (RC1) · Tristan Salles (Referee) · 25 Mar 2020

Barnhart and co-authors present Landlab v2.0 – a community-based Earth surface dynamics model. Their toolkit is shared as an entirely open software on GitHub, allowing for enhanced collaboration and reproducibility of scientific research.

Since the seminal work presented in Hobley et al. 2017 (Landlab v1.0), a growing number of "teachers-students", "users", "user-developers" have started working with the code. The uptake by the community has promoted the development of several new modules ("components") allowing to study a variety of core surface processes unmatched by any other equivalent tools in Earth surface science.

Motivated by (1) capabilities extension (i.e., model grids, utilities, components), (2)

[Figure]

users experience and feedbacks and (3) compatibility with required Python libraries (e.g., decommissioning of Python 2), this communication comprehensively describe the new and major changes that have been undertaken between v1.0 and v2.0 in a way that mirrors the clean and well-written code base.

In addition to the technical section of the paper, I really enjoyed reading the section 5 which I think is a must read for any developer/user looking at building, improving working with scientific codes. The amount of work behind the thorough inclusion of docstrings, unit tests, automatic documentation and integration to the wider range of CSDMS suite of tools is for sure a difficult task that is tremendously time-consuming.

I congratulate the authors and all Landlab contributors for providing the community with such infrastructure. In my opinion the article is ready for publication. And I strongly recommend acceptance of this short communication to the journal.

---

## Referee Comment (RC2) · Wolfgang Schwanghart (Referee) · 27 Mar 2020

In their paper, Barnhart et al. present Landlab 2.0, a new release of the popular numerical environment for landscape evolution modelling. First of all, I like to acknowledge the great tool that Landlab's developers provide to the geomorphology community, and I think it is important that there are papers that accompany the software to make it accessible and citable. ESURF seems to be a perfect outlet for such a paper.

The presented manuscript covers several aspects of software writing by geoscientists and how these aspects pertain to the development of Landlab 2.0. As such, the text is interesting also for a broad readership. However, I think that section 4 is, at least in part, getting very specific and may be sometimes difficult to follow (and potentially

not very interesting) if readers haven't used Landlab before. In fact, the numbered or bulleted lists are very technical and could be part of some technical release notes rather than a scientific paper. I suggest that the changes documented by these lists could be described more generally in "plain words" without the heavy use of syntax wording.

The subsection about citing Landlab should become an own section, and not a subsection of section 4. While I understand that v2.0 introduces some new utilities that provide guidance for users about when to cite what, I think that the issue of citing software is not so much about the release of version 2, but rather a general issue.

Section 5 documents a number of lessons learnt that are pertinent to geoscientific software in general, highlighting the value of unit-testing and the importance of exchange between developers and users.

All in all, I really like the paper. It is well written and interesting to readers, irrespective whether they have used Landlab before or not. My only concern is that the paper mixes quite generic issues in software development in the geosciences with very technical issues specific to Landlab. I suggest to partly rewrite section 4 at those places (bulleted or numbered lists) that read like release notes. In addition, I think that the paper could be equipped with some figures that show some of the output of Landlab, even if it is for mere illustrational purpose.

---

## Author Comment (AC1) · 22 Apr 2020

**1 General updates made independent of reviewer comments**

The manuscript was updated to reflect the addition of one new component.

**2 Response to reviewer Tristan Salles**

We thank reviewer Salles for his review. As he recommended no changes be made, we have made no changes in response to his review.

[Figure]

**3  Response to reviewer Wolfgang Schwanghart**

Reviewer Schwanghart's review recommended that we revise the text to address the following issues:

1. Reduce the use of syntax and bulleted lists, and use more plain language in Section 4 in order to make it more accessible.

2. Add one or more figures that illustrate example output of Landlab.

3. Change the subsection on citing Landlab to its own section.

We agree with all of reviewer Schwanghart's recommendations and have revised the text accordingly. A new Figure 1 provides six example applications from the literature and the subsection on citations is now its own section.

Our revision in response to Point 1 aims to balance our efforts to increase readability and understandability with the value of the technical aspects of the manuscript. We think there is value in providing the technical detail, and in exposing some of the thinking that underlies technical choices. We agree, however, that is is not useful or valuable if it is not understandable.

In our revision of this section, we have focused on improving accessibility wherever possible, and adding context for why such technical text is present. Where possible we worked to connect why technical details or changes matter for an end user. For example, we added background and context on what an interface standard is and how it is useful to an end user to the start of Section 4.2 (Updates to the Component Standard Interface). While we have not excised all use of inline syntax and bulleted lists in the text we have removed much of it. We think our revision of Section 4 has made it more readable and accessible.

**ESurfD**
* * *
Interactive
comment

We note that we have intentionally not made substantial changes to Section 4.3 (Removed or Modified Components and Utilities). One might argue that this section is reminiscent of a change log, yet in many ways that is its intent. This section is designed to highlight major changes and removals *and provide a description of why they were removed*.

In his review Reviewer Schwanghart states "My only concern is that the paper mixes quite generic issues in software development in the geosciences with very technical issues specific to Landlab." This comment is perceptive and clearly identifies two (potentially competing) goals that we had in drafting this manuscript: to document and describe technical details of Landlab *and* to share general lessons learned in the development of Landlab. Yet as the Reviewer Schwanghart points out, these joint goals yield a varyingly technical manuscript. We think that through revisions in response to reviewer comments we have come closer to accomplishing these joint goals.

**Note: As per journal instruction we have not submitted the revised manuscript or a tracked changes manuscript as part of this comment.**

---

## Author Response (AR2)

**Author's Response**

**Short communication: Landlab v2.0: A software package for Earth surface dynamics**

Barnhart et al., submitted 2020

**1  Response to Associate Editor's comments**

We agree that addressing all comments made by Associate Editor Simon Mudd will improve the manuscript. A document with changes tracked follows this page.

**2  Additional Changes**

Lead author Barnhart's affiliation has changed and thus a new affiliation was added.

The details of licenses and attribution are now fully specified for Figure 1.

[revised manuscript text omitted]